# Modeling-Based Risks Assessment and Management of Climate Change in South Korean Forests

**Mina Hong** [1,2] , **Cholho Song** [2] , **Moonil Kim** [3] , **Jiwon Kim** [1] , **Minwoo Roh** [1] , **Youngjin Ko** [1] , **Kijong Cho** [1] , **Yowhan Son** [1] , **Seongwoo Jeon** [1] , **Florian Kraxner** [4] and **Woo-Kyun Lee** [2,*]

1   Department of Environmental Science and Ecological Engineering, Korea University,
    Seoul 02841, Republic of Korea; alsdk920902@korea.ac.kr (M.H.); jiwonandonly@korea.ac.kr (J.K.);
    mw0406toto@korea.ac.kr (M.R.); ko871004@gmail.com (Y.K.); kjcho@korea.ac.kr (K.C.);
    yson@korea.ac.kr (Y.S.); eepps_korea@korea.ac.kr (S.J.)
2   OJEong Resilience Institute (OJERI), Korea University, Seoul 02841, Republic of Korea;
    cholhosong@korea.ac.kr
3   Department of ICT-Integrated Environment, Pyeongtaek University, Pyeongtaek 17869, Republic of Korea;
    futuring.kim@ptu.ac.kr
4   Agriculture Forestry and Ecosystem Services (AFE) Group, Biodiversity and Natural Resources (BNR)
    Program, International Institute for Applied Systems Analysis (IIASA), Schlossplatz 1,
    A-2361 Laxenburg, Austria; kraxner@iiasa.ac.at
*   Correspondence: leewk@korea.ac.kr; Tel.: +82-2-3290-3016

**Abstract:** The IPCC emphasizes the role of forests in the sequestration of greenhouse gases, a significant cause of climate change. Accordingly, it shows the importance of predicting changes in forests due to climate change, evaluating them to reduce vulnerability under adaptive capacity, and finding ways to find climate resilient development pathways. In this study, the KO-G-Dynamic model, a Korean growth model, was linked with the frameworks of AR5 and 6 to assess risk dynamics in the forest growth sector. At this time, the sensitivity is a variability due to the reduction in forest growth, the exposure is the forest as an object, the hazard is climate change, the adaptive capacity is forest management, and the vulnerability is a mechanism that sensitivity could not be adjusted according to adaptive capacity. The risk was assessed by ranking overall risks derived from the process of vulnerability generated by the interaction of the above factors. As a result, the current forests in Korea are age class imbalanced, and the effects of distribution are centered on fast-growing tree species. If climate change and overprotection continue, the vulnerable area expands as sensitivity increases, since the total growth reduces due to increasing over-matured forests. From the regional-based analysis, Gangwon-do and Gyeongsangnam-do mostly consist of the higher V age class, the ratio of 'very high' risk grade was high and the area of 'high' risk grade changed rapidly. However, after applying forest management scenarios of adaptive capacity such as harvesting, reforestation, and thinning based on Republic of Korea's forest management policy, the ratio of 'Low' risk grades increased according to the reduction of vulnerability areas. Therefore, forest management can act as an important factor to reduce the risk of forest growth in response to climate change.

**Keywords:** forest risk assessment; forest growth; climate change; adaptation; forest management

## 1. Introduction

Climate change is a global issue affecting all aspects of human lives, including the surrounding environment and socioeconomic sectors [1]. Under the various complex effects of climate change, the Intergovernmental Panel on Climate Change (IPCC) has emphasized the assessment of future forecasts to prevent increasing risk and vulnerability under limited adaptation capacity [2–4]. However, the complex conceptual approaches in the IPCC Assessment Reports (AR) hinder the development of proper assessment methods in many countries. The previous assessment schemes from AR3 and AR4 focused on vulnerability,

which was calculated by the dynamic combinations of exposure, sensitivity, and adaptation capacity. In contrast, with the growing importance of risk, criteria such as hazard, exposure, and vulnerability have been highlighted as the core of various effects of climate change in AR5.

In AR6, the major concept of AR5 was maintained. However, climate-resilient development pathways (CRDPs), which reflect different and diverse options of adaptive capacities, are emphasized, and the application process of each criterion corresponding to the climate scenarios and modeling mechanism is becoming important [5]. Further assessment is needed to cover the surrounding circumstances of the exposed targets, including climatic variations, expected socio-economic changes, and political instruments for adaptation. In addition, the sensitivity of each criterion and the initial and residual impacts of climate change should be considered. This conceptual complexity leaves room for different interpretations and applications in the assessment process. Furthermore, the modeling process also needs to be modified to represent the impact of climate change and reflect adaptive measures.

According to the Republic of Korea's climate change adaptation policy, each local government sets its climate change adaptation plan based on the Korean National Adaptation Plans (NAP) for climate change [6]. During the first NAP period from 2011 to 2015, the government focused on finding key adaptation problems and developing sectoral planning through qualitative information about the vulnerability. However, only qualitative assessment and normalization that were not reflected in the sensitivity of the mechanism were performed in this step. In addition, it is difficult to evaluate the quantitative future situation under the AR3 and AR4 frameworks [7]. There were limitations in controlling the actual adaptive capacity using adaptability indicators and indicators of exposure, sensitivity, and adaptive capacity in the vulnerability assessment as general and ordinary indicators, depending on the difficulty of sharing interrelationships. To find more accurate adaptation measures to climate change, the NAP must predict quantitative changes. As the second NAP period from 2016 to 2020 began [8], the Republic of Korea's government focused on preparing scientific measures based on quantitative climate change impacts and risk assessments at the national level. Accordingly, to develop tools for evaluating vulnerability and risk, the integrated model for climate change impact and vulnerability assessment (MOTIVE) project and the Vulnerability Assessment Tool to Build Climate Change Adaptation Plan (VESTAP) were established [9,10].

The MOTIVE project and concepts developed a model combining sectors (e.g., health, agriculture, forests, ecology, and oceans) considering the characteristics of Republic of Korea. They were the basis for establishing realistic climate change adaptation measures for the national and local governments. Numerical models have been broadly applied to build key adaptation strategies and to set up budget expenses under the AR5 framework [11]. Furthermore, the recently announced third NAP from 2021 to 2025 was planned to enhance adaptation measures and support regional risk and vulnerability assessments by supporting the existing secondary policy [12]. This was advanced from AR5 and reflected the AR6 concept based on the future social economy.

Forest ecosystems, one of the assessment targets, have been negatively affected by climate change. In the Republic of Korea, the forest growth of pine tree stands is decreasing due to climate warming and frequent forest disasters [13]. Most coniferous forests growing in Republic of Korea are vulnerable to climate change, and the decrease in growth is accelerating, with the mortality of subalpine coniferous forests being particularly prominent [14]. In addition, forest age classes are getting older, and many forest stands are reaching their cutting age. This means Korean forests require proper management and climate change predictions [15]. Forests are also important global carbon sinks [16]. Previous studies evaluated Republic of Korea's forest ecosystem through global models [17–19]. However, mechanical and numerical risk assessments have not comprehensively considered forest management plans and practices in Republic of Korea. Therefore, assessing climate change risk requires proper spatiotemporal models that reflect conceptual linkages among criteria.

In addition, it is important to reduce the risk in forests through adaptation options that reflect the impact of forests and socioeconomic conditions in response to climate change [20]. Therefore, modeling that can reduce risks by encompassing the conceptual frameworks of AR5 and AR6 along with various forest management scenarios is required. The potential risk class and extent of adaptive capacity should be measured according to the mitigation based on forest management scenarios.

In this situation, this study aims to estimate the risks faced by the national and local governments using the AR5 and AR6 frameworks and the Korean forest growth model. Based on this, future risk reduction and scientific-based adaptation directions are suggested.

## 2. Materials and Methods

### 2.1. Study Area

The study area is the forests of the Republic of Korea in longitude from 124°54' to 131°6' and in latitude from 33°9' to 38°45' (Figure 1). Currently, forests in Republic of Korea account for about 62.7% (6,286,438 ha) of the national land, while coniferous forest, broad-leaved forest, and mixed forest account, respectively, for 38.7%, 33.5%, and 27.8% of forest land [21–23]. These forests experienced rapid forest degradation due to the war from 1950 to 1953, but the government-led erosion control and greening project succeeded in forest rehabilitation from 1973 to 1987 [24]. As a result of recovery, the stocking stem volume was 11.31 $m^3$ $ha^{-1}$ at the beginning of the project but increased to 165 $m^3$ $ha^{-1}$ in 2020 [25]. However, the fast-growing tree species planted at this point have recently been classified as vulnerable species to climate change, and forest health problems are becoming an issue due to the unbalanced structure of age class distribution, which accounts for 76% of IV and V age class of forest land [26]. Accordingly, the government is strengthening policies to promote forest management, such as the 6th forest basic planning. Therefore, this study aims to analyze the risks of climate change and suggest efficient countermeasures.

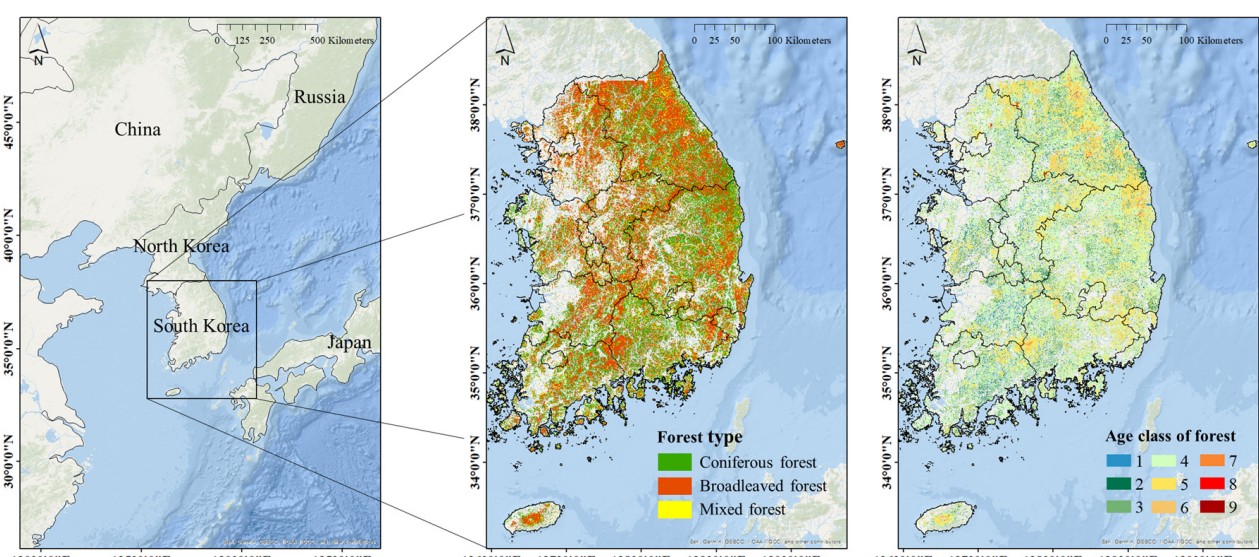

**Figure 1.** Study area and its map of forest type (2018) from Korea Forest Service.

### 2.2. Risk Assessment in Forest Sector

2.2.1. Basic Framework to Link AR5 and AR6 Concept and Modeling Process for Analysis

AR5 and AR6 define risk as being generated by the interaction of climate-related hazards with their levels of exposure, vulnerability, and adaptation (Equation (1)) [27,28]. There were changes from vulnerability to risk, which are the main assessment targets for developing adaptation policies. These differences are reflected in the transformed equations for AR4 and AR5.

$$\text{Risk} = \text{f}(\text{Exposure}, \text{Hazard}, \text{Vulnerability}) \tag{1}$$

$$\text{Vulnerability} = \text{f}(\text{Exposure}, \text{Sensitivity}, \text{Adaptive Capacity}) \qquad (2)$$

These transformations were observed in the changing conceptual process and terms. In AR3 and AR4, sensitivity is the degree to which the system is adversely or beneficially affected by climate-related stimuli; exposure is the degree to which the system is exposed to climate-related stimuli; and adaptive capacity is divided into the extent that the system regulates climate change, mitigates damage, or uses the opportunity to cope with climate change impacts [27,29]. As the AR5 equation was transformed, the existing framework dissolved the new conceptual framework. In the case of sensitivity, the existing designation word disappears, but it improves the mechanism of the exposed targets and the degree of vulnerability through control. Exposure to existing climate change became the subject of current exposure. Sensitivity was defined as the physical phenomenon caused by natural factors, including terms of development such as hazard and organic relation. Vulnerability was defined as a tendency to be negatively influenced (Equation (2)), so the terms were somewhat changed [30,31]. In modeling aspects, these conceptualizations were utilized by proper selection of the model and linkage of input data (Table 1).

**Table 1.** Definitions and utilization of the risk approaches.

| Term | Conception | Definition | Existing Utilization |
|---|---|---|---|
| Sensitivity | AR3, 4 | The degree to which a system is affected, either adversely or beneficially, by climate variability or climate change [32] | Ecosystem mechanism, climate reaction, and inherent processes [33] |
| Exposure | AR3, 4 | The degree to which the system is exposed to climate-related stimuli [34] | Extreme climate [35] |
| | AR5, 6 | The presence of people, livelihoods, species or ecosystems, environmental functions, services and resources, infrastructure, or economic, social, or cultural assets in places and settings that could be adversely affected [36] | Forest ecosystem [37], especially the forest growth model itself [38] |
| Adaptation | AR3, 4, 5, 6 | Initiatives and measures to reduce the vulnerability of natural and human systems against actual or expected climate change effects [39,40] | Anticipatory and reactive, private and public, and autonomous and planned, especially forest management plans and current practices [41,42] |
| Hazard | AR5, 6 | The potential occurrence of a natural or human-induced physical event or trend or physical impact that may cause loss of life, injury, or other health impacts, as well as damage and loss to property, infrastructure, livelihoods, service provision, ecosystems, and environmental resources [34,43,44] | Climate change scenarios such as temperature and precipitation from Representative Concentration Pathways (RCPs) [45,46] |
| Vulnerability | AR3, 4, 5, 6 | The propensity or predisposition to be adversely affected. Vulnerability encompasses a variety of concepts and elements including sensitivity or susceptibility to harm and lack of capacity to cope and adapt [34,47] | The potential negative changes, area, and portion under adaptive measures [48,49] |
| Risk | AR5, 6 | In the context of climate change responses, risks result from the potential of such responses to not achieve the intended objective(s) or potential trade-offs or negative side-effects [5]. Combine the hazard, exposure, and vulnerability to quantify and classify the potential consequences of the risk to the area of investigation. Probabilistic or relative/quantitative terms can be expressed [50] | Combinations of hazard, vulnerability, and adaptation, especially considering probability or different risk management scenarios [51] |

The selection criteria for risk assessment in the forest sector are shown in Figure 2. Forest risk assessment defines the potential reduction of growing stock and tree increment in the country's overall forests as exposure to future climate change and adaptation levels. Thus, in this study, to implement the risk assessment of model-based AR5 and AR6 based on the qualitative vulnerability assessment applied in past research, the hazard defined as a climate-related physical phenomenon can represent the climatic element. Therefore, we utilized the RCP8.5 scenario of future climate change, reflecting time-series changes in temperature and precipitation including extreme climatic and meteorological events [52]. Exposure is where the environmental function is a negatively affected object, including forests. Sensitivity is the degree to which forests are affected by climate change, which is a change in growth [53]. Therefore, although there is a positive relationship between the annual growth and the adjustable sensitivity of each scenario considering seven species of trees (*Pinus densiflora*, *P. koraiensis*, *Larix kaempferi*, *Quercus variabilis*, *Q. mongolica*, Mixed forest A: *P. densiflora* and *Q. variabilis*, and Mixed forest B: *P. densiflora* and *Q. mongolica*), uncontrolled sensitivity is classified as vulnerability, and adaptation should be improved by forest management prescription depending on the vulnerable area. Based on this sensitivity, the vulnerability area was calculated based on the variability in the growth decrease. The greater the variability of growth decrease over time, the higher the vulnerability [17]. The variability of forest ecosystem functions was estimated using the average values for the study period and the entire period (2010–2100) [54]. The potential for adverse consequences of the total risk opportunity derived from the process of vulnerability that cannot be controlled, and controlled sensitivity, can be calculated by risk assessment and quantitatively calculated impact by dynamic interactions among climate-related and adaptive pathways, such as policy and infrastructure [55,56].

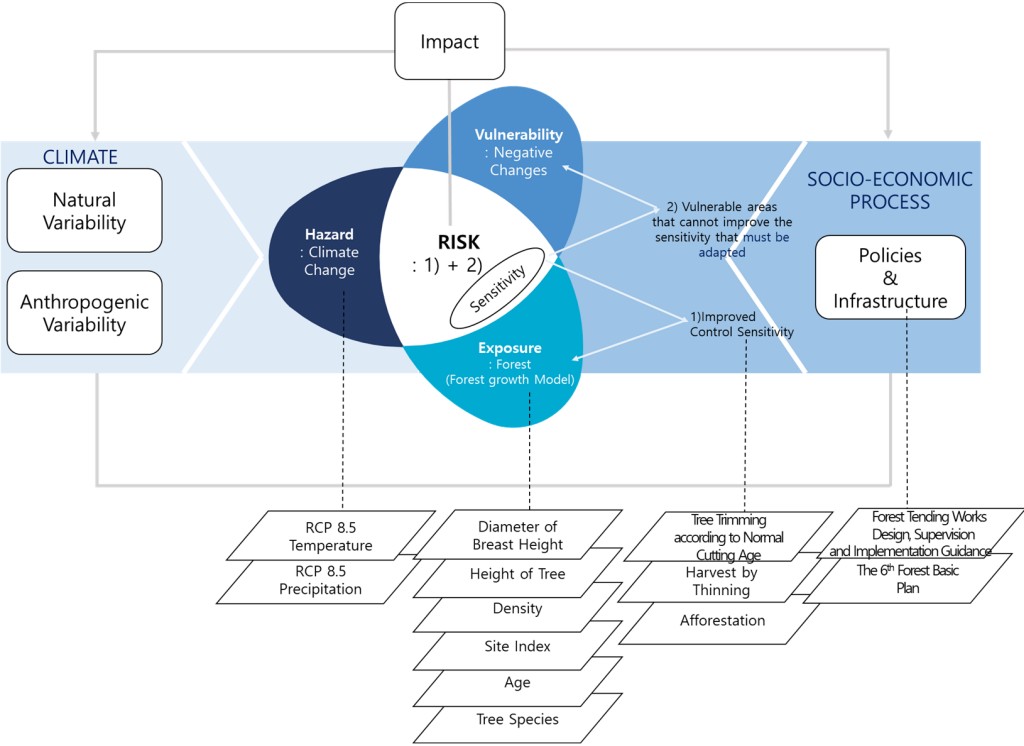

**Figure 2.** Modified risk assessment based on the modeling of the forest growth sector [27,28].

### 2.2.2. Forest Growth Model

The 'KO-G-Dynamic model', a forest growth model, was used to apply risk assessment modeling techniques in this study [57]. Using the KO-G-Dynamic model has enhanced the forest growth of previous studies [58–61]. We constructed the forest stands in 1 km × 1 km grids and described the state of each grid with four variables with varying values: forest

type, site index, stand age, and management type using the fifth NFI and the fifth Forest Cover Map. In addition, each grid was spatially linked to climatic and topographic conditions from digital elevation model data and climate data [62]. Coefficients determined for each tree species were incorporated with the biomass allometric equation data developed by the National Institute of Forest Science (NIFoS) [63]. The KO- can predict the annual growth of temperate forests more accurately, including the climate impact overlooked by traditional dynamic growth models. It can enhance the model's performance, reducing uncertainty about future estimates due to climate change, as it reflects Korea's ecological and environmental characteristics. This can be used to derive growing stock using information such as diameter at breast height, tree height, and stand density [64]. Furthermore, the KO-G-Dynamic model was compared and calibrated through the fifth and sixth National Forest Inventory (NFI) Data and verified through national forestry statistical data and other research [57].

### 2.2.3. Simulating Scenarios

In the study, the visual range of the data was 2030 (2021–2030), 2050 (2041–2050), 2080 (2071–2080), and 2100 (2091–2100) based on current data (2010–2020). This study analyzed risk from the perspective of various scenarios. It estimated the degree to which Republic of Korea's forests change from young managed stand to over-matured forest and become vulnerable according to climate change, and analyzed each scenario to calculate risks through a step-by-step adaptive pathway based on continuous forest management policies.

Scenarios 1 and 2 were developed for the purpose of analyzing the risk of exposed forests based on the hazard degrees. In scenario 1, it was assumed that the current climate is maintained without climate change and that cutting for harvesting is not implemented due to overprotection. Scenario 2 was set to apply the climate change scenario without any cutting (without any adaptation capacity). In scenario 3, climate change scenarios and forest management were applied, considering the legal cutting age and clear-cut harvest area of 15,000 ha per year, which is the current annual harvest area [65]. In scenario 4, climate change scenarios and cutting were also applied, but unlike scenario 3, the clear-cut harvest area was set to be 35,000 ha, which was based on the average stem volume and domestic wood supply in the 6th forest basic planning [66]. Scenarios 3 and 4 were planned to simulate changes in forest physiognomy according to the degree of the adaptive pathway. Furthermore, scenario 3 was constructed to analyze the aspects of Forest and Landscape Restoration (FLR), and scenario 4 to analyze Nature-based Solutions (NBS) [67,68].

After cutting, replanting of suitable species for climate change and thinning intensity were applied equally to each scenario. According to the guidelines for sustainable resource management of the Korea Forest Service, clear-cut is allowed in manageable forest land that has reached normal final age and has a cycle of reforestation [69]. Natural succession and direct land use and land cover change were not included in this study, and it was assumed that forest area remained constant over time.

### 2.3. Data Preparation and Modification

In this study, the risk was evaluated using the dynamic growth model based on quantitative data. The climate data were used as the hazard, the forest type map that could identify the target forest standards as the exposure, and forest management and policy data were used as the adaptive pathway. The 1 km × 1 km of grid-based spatial data were processed, simulated, and evaluated in the model.

The current climate data were from the automated synoptic observing system (2001–2010) from 73 stations, and the climate change data (representative concentration pathways in the HadGEM3ra climate prediction model and the RCP8.5 warming scenario) were detailed for Republic of Korea. The Korea Meteorological Administration provided the high-resolution (1-km spatial resolution) raster-based monthly mean temperature and precipitation data (https://data.kma.go.kr/, accessed on 4 March 2022, see Supplementary Materials). In this study, the provided data were processed and used as annual average climate data.

A forest type map and the fifth and sixth NFI data were used to distinguish seven forest physiognomies considering the characteristics of tree species and upgrade age class information (https://www.bigdata-forest.kr/, accessed on 11 March 2022, see Supplementary Materials). The site index of each stand was estimated based on the height information of the dominant trees in the NFI data and applied to the model. In the forest management segment, manageable areas were classified based on previous research [60]. According to the forest laws and geographical conditions, the available area for harvest was approximately 3,138,000 ha, accounting for about 51.7% of the total forest [62]. In addition, clearcut harvest based on the legal final cutting age assumes that the forest physiognomy reaching the normal final age was fixed, and the forest physiognomy with a high age class and volume was selected as the first to select 15,000 or 35,000 ha. The thinning option was developed to be conducted for grids of approximately 165,000 ha per year, which have reached II and IV, the same as the harvest algorithm. The intensity was determined to conduct approximately 30% of the grids considering the age and volume [69]. A detailed description of the data preparation process is provided in Kim et al., 2019 [57].

## 3. Results

### 3.1. Forest Growth

The sensitivity of forests to hazards was defined as the degree of impact by climate change and estimated through annual growth based on stem volume. The average growth was approximately 4.3 $m^3$ $ha^{-1}$ from 2010 to 2020.

However, the results of each scenario were derived differently depending on the spatiotemporal conditions from 2030 to 2100. In the order of scenarios 1, 2, 3, and 4, the results were 2.98 $m^3$ $ha^{-1}$, 2.88 $m^3$ $ha^{-1}$, 2.88 $m^3$ $ha^{-1}$, and 2.89 $m^3$ $ha^{-1}$ in 2030s; the results were 2.05 $m^3$ $ha^{-1}$, 1.97 $m^3$ $ha^{-1}$, 1.99 $m^3$ $ha^{-1}$, and 2.3 $m^3$ $ha^{-1}$ in 2050s; the results were 1.84 $m^3$ $ha^{-1}$, 1.34 $m^3$ $ha^{-1}$, 1.86 $m^3$ $ha^{-1}$, and 2.1 $m^3$ $ha^{-1}$ in 2080s; and the results were 0.94 $m^3$ $ha^{-1}$, 0.88 $m^3$ $ha^{-1}$, 1.41 $m^3$ $ha^{-1}$, and 1.96 $m^3$ $ha^{-1}$ in 2100s (Figure 3).

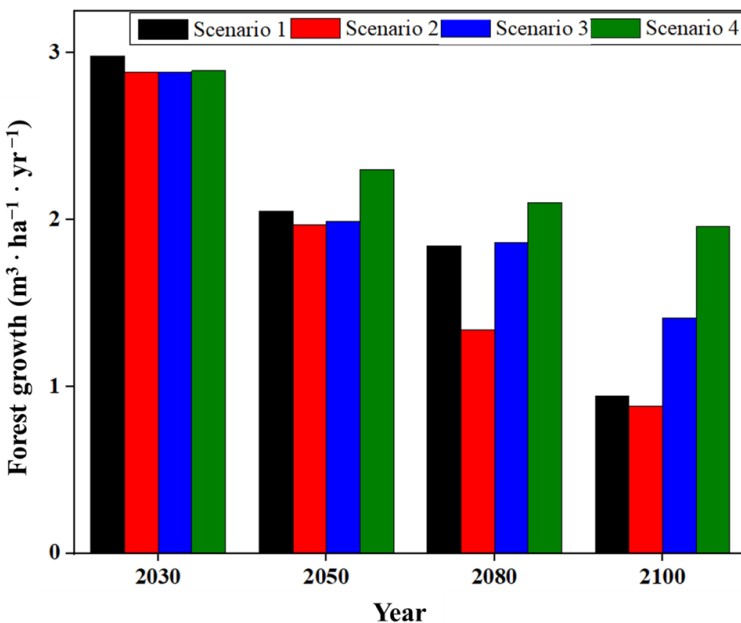

**Figure 3.** Forest growth according to the scenario from 2030 to 2100.

Therefore, comparing scenarios 1 and 2, forest growth was predicted to decrease according to the hazard. Comparing the levels of adaptive pathway in scenario 3 and 4, the higher the advanced management, the higher the forest growth and the lower the sensitivity. Additionally, its impact was increased from a mid-term to long-term perspective.

*3.2. Risk Assessment*

3.2.1. Hazard: Climate Change

A hazard is a physical phenomenon caused by natural factors. As a result of analyzing the annual average temperature and precipitation as a hazard, it was found that the hazard generally increases over time. The average annual temperature increased from 11.51 °C in the 2030s to 15.31 °C in the 2100s. In the case of annual precipitation, it increased from 1430.07 mm in the 2030s to 1600.68 mm in the 2100s (Figure 4). Therefore, the hazard increased by 3.8 °C and 170.61 mm over approximately 80 years.

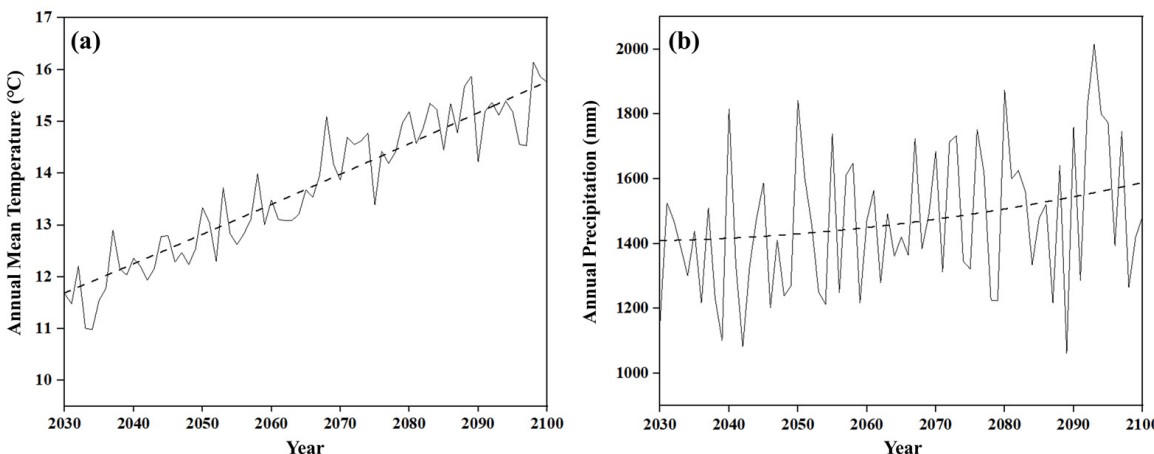

**Figure 4.** The trends of South Korean hazards based on predicted climate change data for 2030–2100. (**a**) Annual Mean Temperature (°C), (**b**) Annual Precipitation (mm).

Regarding temperature, the South Korean forest consists of northern temperate forests (5–10 °C), central temperate forests (10–12 °C), southern temperate forests (12–14 °C), and warm temperate forests (14 °C or higher). However, the classified areas and distribution locations are predicted to change due to climate change. Northern temperate forests accounted for approximately 28% of the forestland in 2020, but it was analyzed that they will account for approximately 8% in the 2050s. Warm temperate forests represented approximately 4% of the forest land in 2020, but the model predicted this to be 61% in the 2080s. It was found that the area of northern temperate forests decreased rapidly from the 2030s because of the hazards and more than half of the forest area became warm temperate forests after the 2080s.

3.2.2. Exposure: Forest

The forest volume subjected to exposure differed depending on the degree and occurrence of the hazard. Furthermore, it was predicted that the shape of the exposure also changes with the advancement of the adaptive pathway.

The estimated stem volumes were 161.86 m$^3$ ha$^{-1}$, 158.67 m$^3$ ha$^{-1}$, 148.01 m$^3$ ha$^{-1}$, and 140.72 m$^3$ ha$^{-1}$ for scenarios 1, 2, 3, and 4 in the 2030s. In the 2050s, they were 210.93 m$^3$ ha$^{-1}$, 202.90 m$^3$ ha$^{-1}$, 190.55 m$^3$ ha$^{-1}$, and 171.38 m$^3$ ha$^{-1}$ for each scenario. In the 2080s, they were 260.22 m$^3$ ha$^{-1}$, 240.80 m$^3$ ha$^{-1}$, 227.55 m$^3$ ha$^{-1}$, 194.00 m$^3$ ha$^{-1}$; and in the 2100s, they were calculated as 283.17 m$^3$ ha$^{-1}$, 258.82 m$^3$ ha$^{-1}$, 243.63 m$^3$ ha$^{-1}$, and 202.56 m$^3$ ha$^{-1}$ (Figure 5).

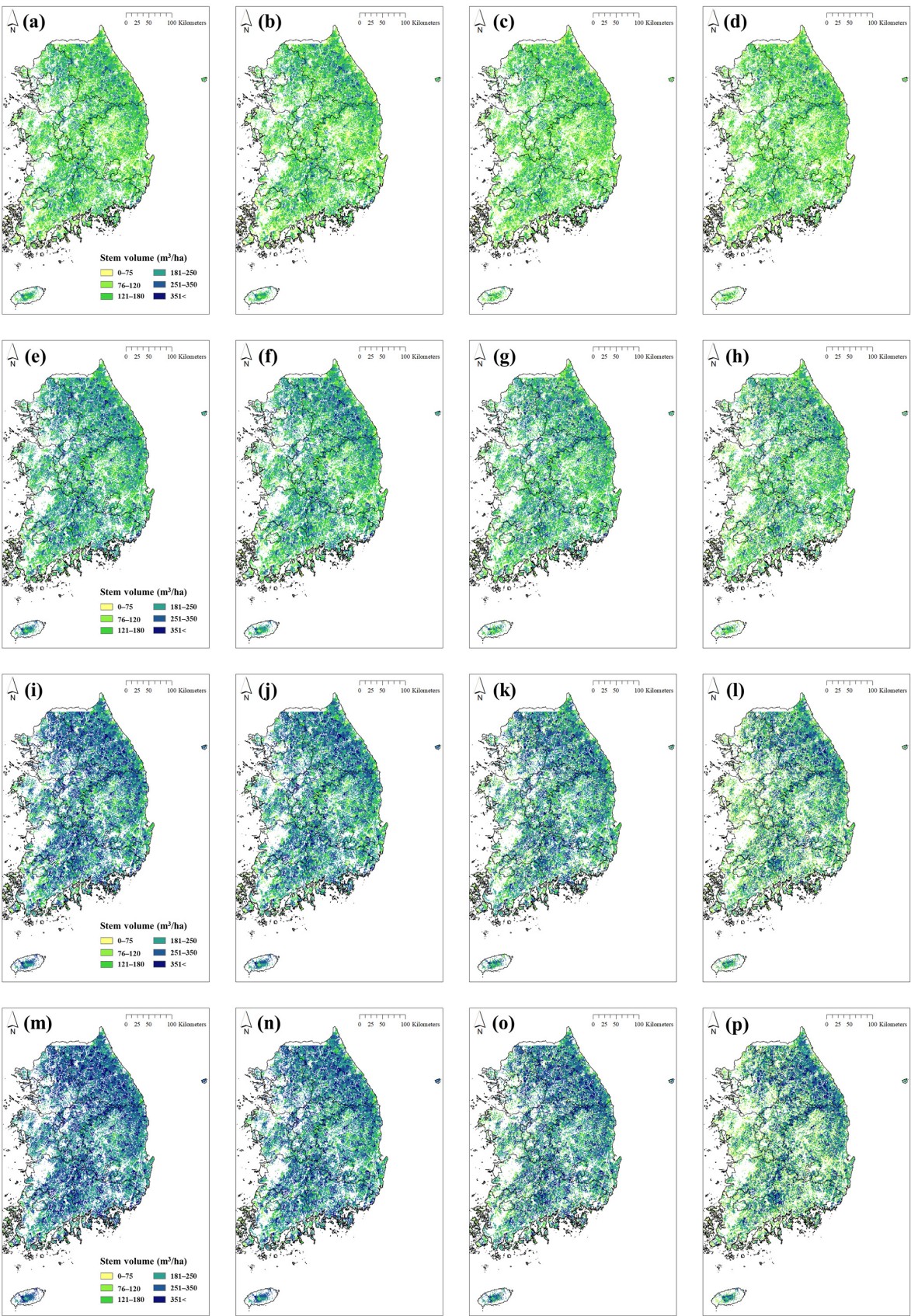

**Figure 5.** Exposure of the forest according to hazard and adaptation. (**a–d**) The stem volume of scenarios 1, 2, 3, and 4 in the 2030s. (**e–h**) The stem volume of scenarios 1, 2, 3, and 4 in the 2050s. (**i–l**) The stem volume of scenarios 1, 2, 3, and 4 in the 2080s. (**m–p**) The stem volume of scenarios 1, 2, 3, and 4 in the 2100s.

Comparing scenarios 1 and 2, the area that was the northern temperate forest close to Gangwon-do showed a slight increase in the stem volume by the 2050s in scenario 2. However, in other areas, the overall mortality of conifers was accelerated, and the stem volume was lower than that in scenario 1. However, in scenarios 3 and 4, the degree of exposure according to the hazard was alleviated with an adaptive pathway, resulting in a balanced stem volume.

### 3.2.3. Adaptive Pathway: Forest Management

Adaptive pathways were analyzed for each spatial distribution based on tree age and stem volume to alleviate sensitivity based on the hazard. Based on the average stem volume and domestic wood supply in the 6th forest basic planning, the climate-resilient development pathways of scenario 3 reflecting the current forest tending and scenario 4 applying the growth maximizing management plan were analyzed.

According to our results, the manageable forest area, excluding the reserve forest, is 3,179,000 ha (52.48% of the forest land). In this case, the rate of clear-cut harvest based on the legal final cutting age and thinning in scenario 3 within the management area from baseline to 2100s was 41.16% and 99.99%, respectively. In scenario 4, it was 95.89% and 99.99%, respectively. In addition, the area rate by the number of thinning based on the management area of scenario 3 was 72.85% in the case of thinning once and 27.14% in the case of thinning twice. In scenario 4, the area for one thinning was 30.70% of the forest management land and 69.27% for thinning twice. Thinning was conducted in most forest lands that could be managed, and it was found that the management area and the number of thinnings increased as the development pathway increased (Figure 6).

The number and intensity of the harvest areas and thinnings were determined based on the age and normal final age by species and volume of the stand. Through the adaptive pathway, climate-resilient development pathways were suggested by improving the age class imbalance and enhancing the health of forests and reforestation species suitable for climate change for the harvest management area.

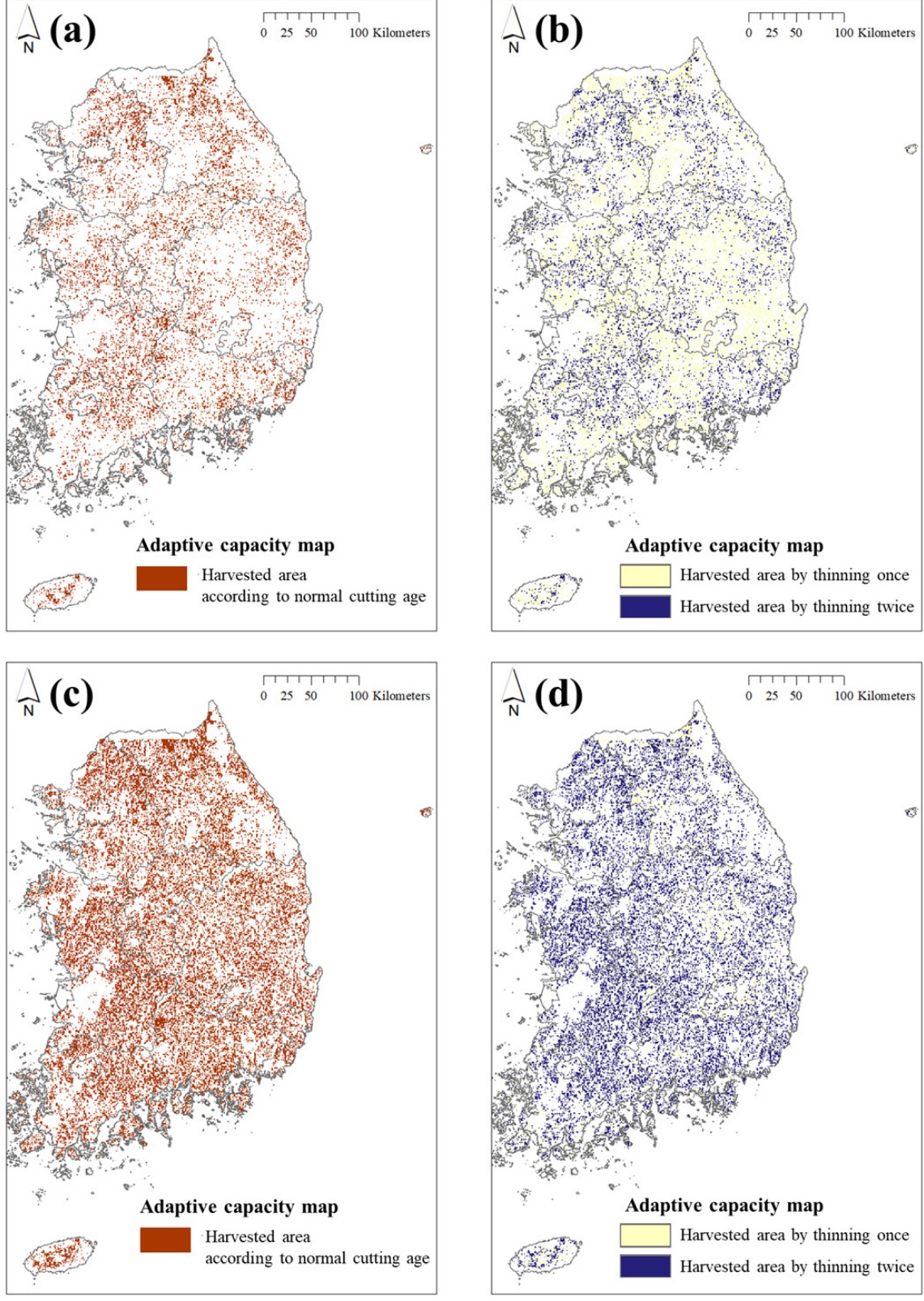

**Figure 6.** The adaptive pathway from baseline to 2100. (**a**) Harvested area according to normal cutting age of scenario 3, (**b**) Harvested area by thinning of scenario 3, (**c**) Harvested area according to normal cutting age of scenario 4, (**d**) Harvested area by thinning of scenario 4.

### 3.2.4. Vulnerability of Forest

Vulnerability refers to the quantitative tendencies that sensitivity and adaptive pathways focus on under changing conditions [70,71]. Therefore, it was predicted that forest growth continues to decrease without management since the hazard or the degree of management is insufficient in forestland. In addition, climate change and a lower degree of management would result in a more extensive vulnerable area.

According to our results, the vulnerable areas of scenarios 1, 2, 3, and 4 in the 2030s were 5,482,500 ha yr$^{-1}$, 5,862,800 ha yr$^{-1}$, 4,548,800 ha yr$^{-1}$, and 4,482,000 ha yr$^{-1}$. They were calculated to be 6,011,700 ha yr$^{-1}$, 6,037,500 ha yr$^{-1}$, 5,784,700 ha yr$^{-1}$, 5,525,900 ha yr$^{-1}$ in the 2050s, and 5,855,400 ha yr$^{-1}$, 6,654,000 ha yr$^{-1}$, 5,462,300 ha yr$^{-1}$, 4,787,100 ha yr$^{-1}$ in the 2080s. In the 2100s, they were 5,460,400 ha yr$^{-1}$, 6,056,400 ha yr$^{-1}$, 5,302,600 ha yr$^{-1}$, and 4,666,800 ha yr$^{-1}$ (Figure 7).

The difference in the degree of vulnerability was evident from the mid- to long-term perspectives rather than the short-term perspective. Particularly, in scenarios 3 and 4, the forest management scenario was applied to improve the structure and health of forests through balanced management of stand density and age class based on cutting and thinning and reforestation of appropriate species to respond to climate change in the managed area. In addition, the degree of vulnerability was lower than that in over-protected scenarios 1 and 2. Furthermore, in scenario 2, in which climate change and overprotection were implemented, most of the forestland was vulnerable from the 2050s onward.

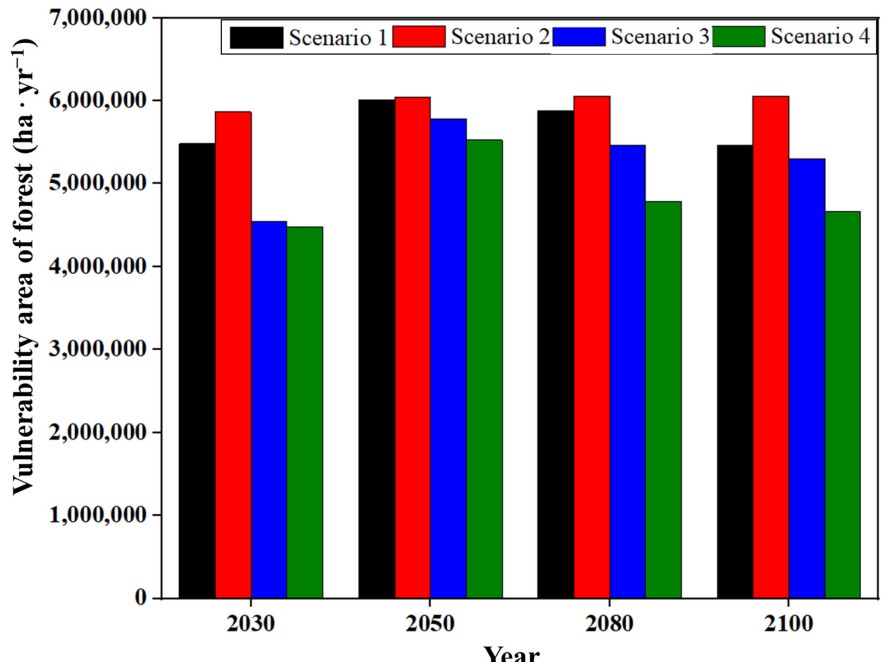

**Figure 7.** Estimated vulnerability area from the 2030s to the 2100s using scenarios 1–4.

### 3.2.5. Risk Assessment of Forest

Forest risk assessment was conducted by analyzing the tendency of climate change and growth based on the previous vulnerability evaluation and grading of the matrix method (very high/high/moderate/low). Risk was determined by how persistent maladaptations, such as reductions in mortality and growth, occur in response to climate change.

As a result of this study, as Republic of Korea's age class progressed from IV to V and was converted into an over-matured forest, climate change increased the risk area. It also increased the very high-grade risk area from 2030 to 2100. However, it was predicted that the risk area would decrease, and the low-grade risk area would increase as the forest management degree, i.e., harvest and reforestation of appropriate species, increased due to climate change (Figure 8).

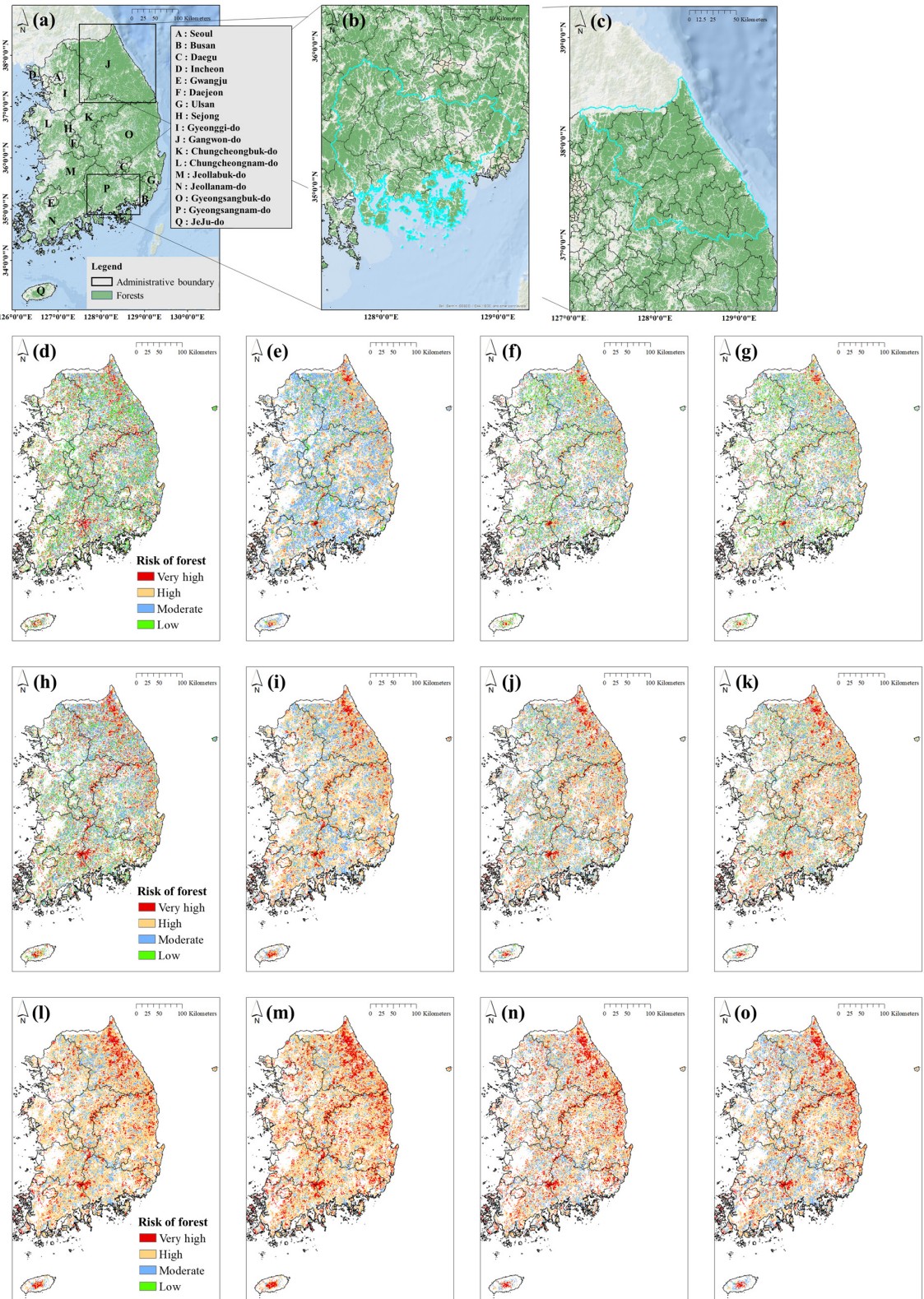

**Figure 8.** Risk assessment map of the forest sector. (**a**) Administrative boundaries and forest distribution of Republic of Korea. (**b**) Administrative boundaries and forest distribution within Gyeongsangnam-do. (**c**) Administrative boundaries and forest distribution within Gangwon-do. (**d–g**) The risk of scenarios 1, 2, 3, and 4 in the 2030s. (**h–k**) The risk of scenarios 1, 2, 3, and 4 in the 2050s. (**l–o**) The risk of scenarios 1, 2, 3, and 4 in the 2080s.

The risk areas based on forest land by region were compared and analyzed to examine this at a regional level. The very high grades of Gangwon-do, Gyeongsangnam-do, Busan, Ulsan, and Jeju-do, which are aging first compared to other regions, had a high proportion of, or change to, high grades, and this occurred rapidly. Accordingly, the ratio of the 2030s with a very high rating in Gangwon-do, a representative region for the reduction of northern temperate forests, was estimated to be 11.95% in scenario 1, 43.34% in scenario 2, 7.11% in scenario 3, and 7.03% in scenario 4. In the 2100s, it accounted for 65.78%, 64.08%, 29.74%, and 28.39%, respectively, and increased in the long term. The low grades were 18.67%, 5.53%, 12.53%, and 12.26%, respectively, in the 2030s, and 0%, 0%, 0.01%, and 2.69% in the 2100s, respectively. Although the low grade is gradually decreasing, it is predicted that its distribution will increase as forest management continues.

In the case of Gyeongsangnam-do, a representative of the transition to warm temperate forests, the very high grades were 14.98%, 3.31%, 7.94%, and 7.87%, respectively, in the 2030s. The overall trend increased to 24.37%, 3.09%, 12.08%, and 14.73% in the 2100s. However, although scenario 4 with ideal management was estimated to be somewhat higher, considering the overall risk, the total risk area of Gyeongsangnam-do in 2100 was 4943 ha in scenario 3 and 4894 ha in scenario 4, indicating that the risk area of scenario 4 was smaller than that of scenario 3. In addition, the rate of the low-grade area was 24.37%, 3.09%, 12.08%, and 14.73% in the 2030s, respectively, and decreased to 0%, 0%, 0%, and 2.15% in the 2100s. However, the extent of the decrease was reduced with the adaptive pathway for advanced management (Figure 9).

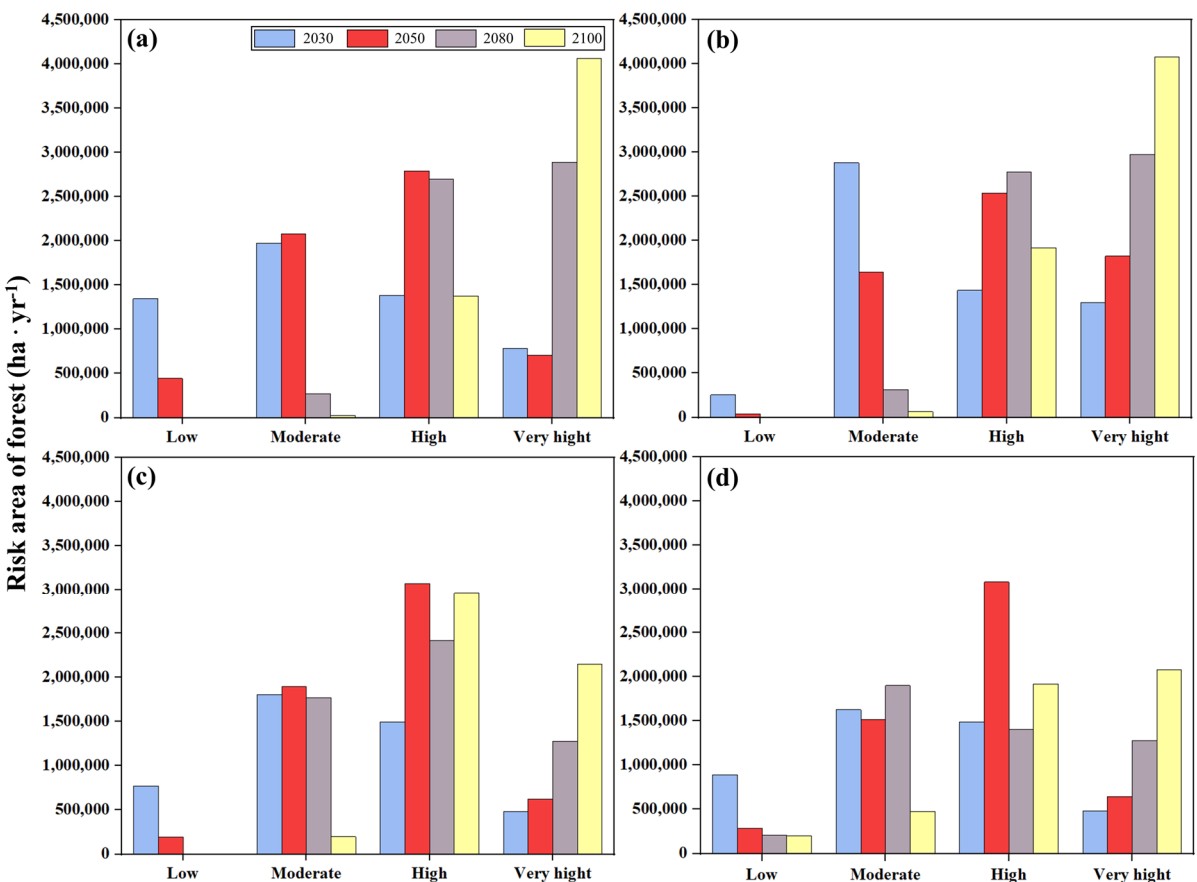

**Figure 9.** Risk grade areas from 2030 to 2100. (**a**) Risk area of scenario 1. (**b**) Risk area of scenario 2. (**c**) Risk area of scenario 3. (**d**) Risk area of scenario 4.

## 4. Discussion

Assessing the risk of the forest sector due to climate change in the Republic of Korea will help the country recognize the hazards of forest growth and establish adaptive measures [72]. There have been many assessments in previous studies, but studies on the linkage between assessment factors were insufficient [54,73,74]. In this study, the vulnerability and growth-related risks were evaluated to fit the AR5 and AR6 frames based on a validated model with the qualitative and limited adjustment range of the vulnerability assessment covered by AR4. In addition, it is assumed that adaptation factors for sensitivity are collectively implemented throughout the forest physiognomy.

The negative effect on the forest sector by 2100 can be identified as being based on climate change in Republic of Korea, and the risk can be reduced through climate-resilient development pathways according to policies reflecting Korea's environmental and ecological characteristics. South Korean forests are currently unbalanced in age class due to the planting of fast-growing tree species planted during the national erosion control and greening project period [75]. In particular, Gangwon-do is a representative warm temperate forest where the V age class accounted for most of the forest and was most affected by climate change [76]. Accordingly, scenario 2, applying climate change and overprotection, was predicted to have 193,600 ha be more vulnerable from 2030 to 2100. However, in the case of scenario 4, which applied climate resilient development pathway forest management, it was analyzed that about 184,800 ha was more vulnerable in 2100 than in 2030 after gradually adjusting the reforestation of appropriate species according to climate change.

According to the risk assessment, in scenario 2, where overprotection was applied, approximately 68% of the forest was concentrated at a very high level by 2100. On the other hand, in scenario 4, where ideal forest management was applied, it was estimated that only approximately 34% was distributed at a very high level. Therefore, it was estimated that the more the forest management progressed, the smaller the vulnerable area and the lower the risk grade compared to the overprotected scenario. This suggests that forest management including NBS as well as FLR should be more promoted.

In this study, protection was continuously carried out in the forest protection land. The appropriate level of forest management was applied to the forest management land and analyzed according to each purpose [57]. However, this study has limitations because it does not consider natural (e.g., pests and wind) and human-induced disturbances (e.g., forest fire, land cover, and logging). In addition, as the overall risk trend of forest growth is analyzed using climate change scenarios that include extreme climate, there is a limitation of being unable to conduct a short-term risk assessment during extreme climate (e.g., heavy rain, drought). Additionally, this study has limitations of not considering the management plan according to forest ownership (national forest, public forest, private forest). In the case of private forests in Korea, the proportion of mountain owners owning forests of less than 3 ha accounts for 86% [21]. This has the characteristics of extensive forest management [77]. In future research, advanced risk assessments can be made through convergence with overseas models that reflect forest disturbance, extreme climate, and forest ownership [78–82].

This study is meaningful because it evaluated the forest ecosystem due to climate change by applying a Korean forest growth model (KO-G-Dynamic model) that can analyze forests spatiotemporally removed from the vulnerability evaluation according to the indicators. Through this, it is possible to evaluate the current overall forest status and the forest status at the regional level. In particular, the regional risk area quantified by grade can be scientifically based on the degree of local government forest management. In addition, deviating from the previous statistically-based forest management plan [83], the spatiotemporally predicted data of 1 km$^2$ unit can be used to estimate the state of the stand in the future and establish a management plan.

This suggests an optimal climate-resilient development pathway for sustainable forests. Furthermore, this study can contribute to establishing carbon-neutral and sustainable policy

goals in the rapidly changing climate and socioeconomic environment. In addition, as this study confirms that risk assessment of the forest growth sector in Republic of Korea is possible, it suggests that it can be applied to risk assessment of the overseas forest growth sector in the future. Additionally, it can support other countries in establishing international adaptation policies or implementing risk assessments.

## 5. Conclusions

In this study, the risk assessment of the forest sector due to climate change was performed by analyzing time and space data, such as climate change scenarios and forest growth models, in line with the newly revised IPCC AR5 and AR6 frames. The risk assessment was developed using the qualitative data corresponding to the vulnerability assessment of the existing AR4 and adding the model to the verified result. For the risk assessment, the vulnerability was applied to the exposure, hazard, and sensitivity of the mechanism, and adaptive capacity was considered and evaluated. As a result, the potential sensitivity of forest growth is reduced. If the sensitivity is adjusted through proper climate-resilient development pathways for risk reduction, it is possible to prevent a rapid decrease in growth.

Thus, it was concluded that the risk impact was reduced only by reducing vulnerability by recognizing and adjusting the sensitivities of the appropriate mechanisms according to climate change.

Therefore, with the advent of a new climate change regime, national and regional forest management policies must be implemented with clear goals and directions. Furthermore, it is important to quantify forest risk through dynamic forest growth modeling from a scientific perspective and adjust the climate-resilient development pathway accordingly. In the future, it is essential to use this to advance modeling that reflects the factors of diversifying forests to establish and improve laws and policies according to climate change. Consequently, it is expected that a risk assessment technique using this model will be needed in the future.

**Supplementary Materials:** Forest data are available online through the Forest Big Data Exchange of the South Korea website (https://www.bigdata-forest.kr/, accessed on 11 March 2022). Climate data were obtained from the Korea Meteorological Administration (https://data.kma.go.kr/, accessed on 24 March 2022).

**Author Contributions:** Conceptualization, M.H., C.S., M.K. and W.-L.; Data curation, J.K., M.R., C.S. and Y.K.; Formal analysis, M.H.; Methodology, M.H., M.K. and W.L.; Supervision, C.S., M.K. and W.L.; Validation, C.S. and M.K.; Visualization, M.H., M.R. and W.L.; Writing—original draft, M.H.; Writing—review and editing, C.S., M.K., K.C., Y.S., S.J., F.K. and W.L. All authors have read and agreed to the published version of the manuscript.

**Funding:** This work was supported by Korea Environment Industry & Technology Institute (KEITI) through "Climate Change R&D Project for New Climate Regime (RE202201934)", funded by Korea Ministry of Environment (MOE). The study was also carried out with the support of 'R&D Program for Forest Science Technology (Project No. 2021363B10-2223-BD01)' provided by Korea Forest Service (Korea Forestry Promotion Institute).

**Data Availability Statement:** Not applicable.

**Acknowledgments:** The authors appreciate the financial support from the Korea Ministry of Environment and the Korea Forest Service. The authors also feel gratitude to Ji-Sang Lee for computer coding of the model.

**Conflicts of Interest:** The authors declare no conflict of interest.

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
