# Peer review of "Modeling-Based Risks Assessment and Management of Climate Change in South Korean Forests"

_forests, doi:10.3390/f14040745_

Round 1

Reviewer 1 Report

The paper entitled “Risk Assessment of Climate Change in Forest based on Growth 2 Modeling in South Korea” try to evaluate the risk level of forests to climate change in South Korea. This study introduced thinning as an important adaptive pathway for forest management. A forest growth model, “KO-G-Dynamic mode” was applied in the risk assessment. My concern is the definition of hazard in this study. In the part “3.2.1” the authors analyzed the annual changes of temperature and precipitation as the trends of hazards. I wouldn’t say the increasing of temperature and precipitation is a “hazard”. Please clarify how this paper define and express hazard with climate data. By the way, in line 502, the part “acknowledgments” should be rewritten.

Author Response

Answers to Reviewers’ Comments

Title: Risk Assessment of Climate Change in Forest based on Growth Modeling in South Korea.

We appreciate your valuable comments and suggestions; these have helped us to improve upon the quality of our manuscript. The manuscript has been revised in accordance with your suggestions. The revised manuscript has been re-checked by a native English language editor to improve readability. Please find our responses below for each comment (P and L mean page number and line number in “Revised manuscript”). We hope that you find our revised manuscript acceptable.

Comment 1

The paper entitled “Risk Assessment of Climate Change in Forest based on Growth 2 Modeling in South Korea” try to evaluate the risk level of forests to climate change in South Korea. This study introduced thinning as an important adaptive pathway for forest management. A forest growth model, “KO-G-Dynamic mode” was applied in the risk assessment. My concern is the definition of hazard in this study. In the part “3.2.1” the authors analyzed the annual changes of temperature and precipitation as the trends of hazards. I wouldn’t say the increasing of temperature and precipitation is a “hazard”. Please clarify how this paper define and express hazard with climate data. By the way, in line 502, the part “acknowledgments” should be rewritten.

→Response

Thank you for your advice, and we understand your comments based on the definition in Table 1. However, we also need to utilize it following previous documents as well as considering the characteristics of the forest growth, which have been affected by daily changes in temperature and precipitation. Thus, we have defined the concepts (AR3, 4, 5, 6) and utilized it as in Table 1. As you commented, regarding the situation of concern, L158-160 in the methodology and L455-467 in the discussion were supplemented and written.

Furthermore, 3.2.1 is the result part, which refers to the results of meteorological conditions that negatively affect forest growth by referring to existing studies. Therefore, the relevant part was composed of the result part of 3.2.1 in terms of expressing the results according to the factors for risk assessment.

In addition, as you commented, mortality due to extreme climate can be expressed, but this study focuses on the analysis of long-term growth decrease due to climate change, including extreme climate. Finally, based on your kind comments, we've rewritten the "acknowledgments."

(P6 L158-160)

Thus, in this study, to implement the risk assessment of model-based AR5 and AR6 based on the qualitative vulnerability assessment applied in past research, the hazard defined as a climate-related physical phenomenon can represent the climatic element. Therefore, We utilized the RCP8.5 scenario of future climate change reflecting time-series changes in temperature and precipitation including extreme climatic and meteorological events [49].

Added references

  • Van Vuuren, D. P., Riahi, K., Moss, R., Edmonds, J., Thomson, A., Nakicenovic, N., ... & Arnell, N. (2012). A proposal for a new scenario framework to support research and assessment in different climate research communities. Global Environmental Change, 22(1), 21-35.

(P17 L455-467)

In this study, protection was continuously carried out in the forest protection land. The appropriate level of forest management was applied to the forest management land and analyzed according to each purpose [55]. However, this study has limitations because it does not consider natural (e.g., pests and wind) and human-induced disturbances (e.g., forest fire, land cover, and logging). In addition, as the overall risk trend of forest growth is analyzed using climate change scenarios that include extreme climate, there is a limitation in being unable to conduct a short-term risk assessment during extreme climate (e.g., heavy rain, drought). Also, this study has limitations in not considering the management plan according to forest ownership (national forest, public forest, private forest). In the case of private forests in Korea, the proportion of mountain owners owning forests of less than 3ha accounts for 86% [18]. It has the characteristics of extensive forest management [74]. In future research, advanced risk assessments can be made through convergence with overseas models that reflect forest disturbance, extreme climate, and forest ownership [75–79].

(P18 L524-528)

Acknowledgments: The authors appreciate the financial support from the Korea Ministry of Environment and the Korea Forest Service. The authors also feel gratitude to Ji-Sang Lee for computer coding of the model. We deeply feel thankful to the four anonymous reviewers for their comments and critical reading which helped improve and clarify this manuscript greatly.

Reviewer 2 Report

The subject is relevant. The manuscript has written well with decent structure. I think this manuscript can be considered for publication after some modifications.

COMMENTS

- Section 2.2.2 needs more explanation about structure of the model, inputs, outputs, accuracy, previous evaluations, etc. maybe in an extra paragraph. The used model is the core of our study, but it has been explained very briefly.

- Similarly, section 2.3 required more clarification. For instance, what is the source for climate data? They were daily, monthly, or annually? Where were the location for the selected weather stations? etc.

- And the most important issue: was the model calibrated and validated previously? It is very important, because the outputs of a simulation model can be interpreted, only after the model was calibrated and validated.

- Location of weather stations were not presented in any figure. How did you correct the biases in CMIP5 or CMIP6 model? And why did you use CMIP5 climate change scenarios (e.g., RCP8.5), while CMIP6 is available and more reliable for future projections?

Author Response

Answers to Reviewers’ Comments

Title: Risk Assessment of Climate Change in Forest based on Growth Modeling in South Korea.

We appreciate your valuable comments and suggestions; these have helped us to improve upon the quality of our manuscript. The manuscript has been revised in accordance with your suggestions. The revised manuscript has been re-checked by a native English language editor to improve readability. Please find our responses below for each comment (P and L mean page number and line number in “Revised manuscript”). We hope that you find our revised manuscript acceptable.

Comment 1

Section 2.2.2 needs more explanation about structure of the model, inputs, outputs, accuracy, previous evaluations, etc. maybe in an extra paragraph. The used model is the core of our study, but it has been explained very briefly.

→Response

Thank you for your advice and suggestion. We have added some related references and sentences in L190-199. Also, the part about this model's more detailed input and output data was written in 2.3.

(P7 L190-199)

It can enhance the model's performance, reducing uncertainty about future estimates due to climate change as it reflects Korea's ecological and environmental characteristics. This can be used to derive growing stock using information such as diameter at breast height, tree height, and stand density [60]. Furthermore, the KO-G-Dynamic model was compared and calibrated through the fifth and sixth National Forest Inventory (NFI) Data and verified through national forestry statistical data and other research [55].

Added references

  • Kim, M., Kraxner, F., Son, Y., Jeon, S. W., Shvidenko, A., Schepaschenko, D., ... & Lee, W. K. (2019). Quantifying impacts of national-scale afforestation on carbon budgets in South Korea from 1961 to 2014. Forests, 10(7), 579.
  • Hong, M., Song, C., Kim, M., Kim, J., Lee, S. G., Lim, C. H., ... & Lee, W. K. (2022). Application of integrated Korean forest growth dynamics model to meet NDC target by considering forest management scenarios and budget. Carbon Balance and Management, 17(1), 1-18.

(P7-8)

2.3. Data preparation and modification

In this study, the risk was evaluated using the dynamic growth model based on quantitative data. The climate data was used as the hazard, the forest type map that could identify the target forest standards as the exposure, and forest management and policy data were used as the adaptive pathway. The 1 km × 1 km of grid-based spatial data was processed, simulated and evaluated in the model.

The current climate data was from the automated synoptic observing system (2001–2010) from 73 stations, and the climate change data (representative concentration pathways in the HadGEM3ra climate prediction model and the RCP8.5 warming scenario) was detailed for South Korea. The Korea Meteorological Administration provided the high-resolution (1-km spatial resolution) raster-based monthly mean temperature and precipitation data (https://data.kma.go.kr/). In this study, the provided data were processed and used as annual average climate data.

A forest type map and the fifth and sixth NFI Data were used to distinguish seven forest physiognomies considering the characteristics of tree species and upgrade age class information. The site index of each stand was estimated based on the height information of the dominant trees in NFI data and applied to the model. In the forest management part, manageable areas were classified based on previous research [58]. According to the forest laws and geographical conditions, the available area for harvest was approximately 3,138,000 ha, accounting for about 51.7% of the total forest [66]. In addition, clear-cut harvest based on the legal final cutting age assumes that the forest physiognomy reaching the normal final age was fixed, and the forest physiognomy with a high age class and volume were selected as the first to select 15,000 or 35,000 ha. Thinning option was developed to be conducted for grids of approximately 165,000 ha per year, where have reached II and IV, as same as the harvest algorithm. The intensity was determined to conduct approximately 30% of them considering the age and volume [65]. A detailed description of the data preparation process is provided in Kim et al., 2019 [55].

Comment 2

Similarly, section 2.3 required more clarification. For instance, what is the source for climate data? They were daily, monthly, or annually? Where were the location for the selected weather stations? etc.

→Response

Thank you for your kind advice and we agreed with your comment. We have added L241-248. This study processed and applied monthly climate data provided by the Korea Meteorological Administration as annual climate data.

(P8 L241-248)

The current climate data was from the automated synoptic observing system (2001–2010) from 73 stations, and the climate change data (representative concentration pathways in the HadGEM3ra climate prediction model and the RCP8.5 warming scenario) was detailed for South Korea. The Korea Meteorological Administration provided the high-resolution (1-km spatial resolution) raster-based monthly mean temperature and precipitation data (https://data.kma.go.kr/). In this study, the provided data were processed and used as annual average climate data.

Comment 3

And the most important issue: was the model calibrated and validated previously? It is very important, because the outputs of a simulation model can be interpreted, only after the model was calibrated and validated.

→Response

Thank you for your kind advice and we agreed with your comment. We have added L190-199 and references. In the previous study by Kim et al., 2019, this model was calibrated through the 5th and 6th NFI and verified through the national forestry statistics and other research (R2 = 0.96, RMSE = 9.13).

(P7 L190-199)

It can enhance the model's performance, reducing uncertainty about future estimates due to climate change as it reflects Korea's ecological and environmental characteristics. This can be used to derive growing stock using information such as diameter at breast height, tree height, and stand density [60]. Furthermore, the KO-G-Dynamic model was compared and calibrated through the fifth and sixth National Forest Inventory (NFI) Data and verified through national forestry statistical data and other research [55].

Added references

  • Kim, M., Kraxner, F., Son, Y., Jeon, S. W., Shvidenko, A., Schepaschenko, D., ... & Lee, W. K. (2019). Quantifying impacts of national-scale afforestation on carbon budgets in South Korea from 1961 to 2014. Forests, 10(7), 579.
  • Hong, M., Song, C., Kim, M., Kim, J., Lee, S. G., Lim, C. H., ... & Lee, W. K. (2022). Application of integrated Korean forest growth dynamics model to meet NDC target by considering forest management scenarios and budget. Carbon Balance and Management, 17(1), 1-18.

Comment 4

Location of weather stations were not presented in any figure. How did you correct the biases in CMIP5 or CMIP6 model? And why did you use CMIP5 climate change scenarios (e.g., RCP8.5), while CMIP6 is available and more reliable for future projections?

→Response

Apologies for bringing any questions due to insufficient description of this issue. In the case of meteorological observation data (2000-2010) of the baseline, it was provided based on 73 observation stations. The contents were added to L241-248. In addition, as the concept of risk assessment began to be emphasized from the IPCC AR5 report, this study used RCP8.5 among CMIP5 climate change scenarios as a study to analyze the possibility of risk assessment in South Korea. Also, in the case of RCP 8.5, the scenario was applied as it was possible to analyze the most realistic situation with the current trend of greenhouse gas emissions.

About the CIMP6, some are reliable, and some are not, because they are scenarios. Recently KMA released the downscaled climate data based on SSP scenario for Korea, but available data starts from 2015, so in that case, the established model had a time gap between 2011-2015. Thus, we use CMIP 5 data first to fit the model and to know future trends based on the risk concept. We will also develop model for CMIP6 soon.

(P8 L241-248)

The current climate data was from the automated synoptic observing system (2001–2010) from 73 stations, and the climate change data (representative concentration pathways in the HadGEM3ra climate prediction model and the RCP8.5 warming scenario) was detailed for South Korea. The Korea Meteorological Administration provided the high-resolution (1-km spatial resolution) raster-based monthly mean temperature and precipitation data (https://data.kma.go.kr/). In this study, the provided data were processed and used as annual average climate data.

Reviewer 3 Report

The authors tried to cover an important aspect of the Risks Assessment of climate change in forests using growth modeling in South Korea. The article is well written and there are some minor modifications needed-

1. title must be modified to 'Modeling-based Risks Assessment and Management of Climate Change in South Korean Forests'

2. In line 112: 27.8 must be 27.8%

3. In case of uses of botanical names, please add authority in first-time uses in both abstract and main texts, thereafter without authority. Again two species of the same genus like Pinus densiflora, Pinus koraiensis; please write like Pinus densiflora, P. koraiensis (all botanical names will be italiced and authority will be in normal fonts)

4. Conclusion must be Conclusions

5. Add one para on the caveat of your study at the end of the discussion

6. Also suggested adding one specific para on the risk management options including FLR, NbS to justify your research

Author Response

Answers to Reviewers’ Comments

Title: Risk Assessment of Climate Change in Forest based on Growth Modeling in South Korea.

We appreciate your valuable comments and suggestions; these have helped us to improve upon the quality of our manuscript. The manuscript has been revised in accordance with your suggestions. The revised manuscript has been re-checked by a native English language editor to improve readability. Please find our responses below for each comment (P and L mean page number and line number in “Revised manuscript”). We hope that you find our revised manuscript acceptable.

Comment 1

title must be modified to 'Modeling-based Risks Assessment and Management of Climate Change in South Korean Forests’

→Response

Thank you for your valuable comment on our manuscript. According to your comment, we changed the title to “Modeling-based Risks Assessment and Management of Climate Change in South Korean Forests”.

(P1 Title)

Modeling-based Risks Assessment and Management of Climate Change in South Korean Forests

Comment 2

In line 112: 27.8 must be 27.8%

→Response

Thank you for your detailed comment and we are sorry for the confusion in that sentence. Based on your good comment, we have modified an sentence.

(P3 L112)

Currently, forests in South Korea account for about 62.7% (6,286,438 ha) of the national land, while coniferous forest, broadleaved forest, and mixed forest account respectively for 38.7%, 33.5%, and 27.8% in forest land [18–20].

Comment 3

In case of uses of botanical names, please add authority in first-time uses in both abstract and main texts, thereafter without authority. Again two species of the same genus like Pinus densiflora, Pinus koraiensis; please write like Pinus densiflora, P. koraiensis (all botanical names will be italiced and authority will be in normal fonts)

→Response

Thank you for your kind advice and we agreed with your comment. We have modified L167-169.

(P6 L167-169)

Therefore, although there is a positive relationship between the annual growth and the adjustable sensitivity of each scenario considering seven species of trees (Pinus densiflora, P. koraiensis, Larix kaempferi, Quercus variabilis, Q. mongolica, Mixed forest A: P. densiflora and Q. variabilis, and Mixed forest B: P. densiflora and Q. mongolica), uncontrolled sensitivity is classified as vulnerability, and adaptation should be improved by forest management prescription depending on the vulnerable area.

Comment 4

Conclusion must be Conclusions

→Response

Thanks for your kind comments. Based on your comments, we revised the word.

(P18 L485)

  1. Conclusions

Comment 5

Add one para on the caveat of your study at the end of the discussion

→Response

Thank you for your advice and suggestion. We have added some related references and sentences in L455-468.

(P17 L455-468)

In this study, protection was continuously carried out in the forest protection land. The appropriate level of forest management was applied to the forest management land and analyzed according to each purpose [55]. However, this study has limitations because it does not consider natural (e.g., pests and wind) and human-induced disturbances (e.g., forest fire, land cover, and logging). In addition, as the overall risk trend of forest growth is analyzed using climate change scenarios that include extreme climate, there is a limitation in being unable to conduct a short-term risk assessment during extreme climate (e.g., heavy rain, drought). Also, this study has limitations in not considering the management plan according to forest ownership (national forest, public forest, private forest). In the case of private forests in Korea, the proportion of mountain owners owning forests of less than 3ha accounts for 86% [18]. It has the characteristics of extensive forest management [74]. In future research, advanced risk assessments can be made through convergence with overseas models that reflect forest disturbance, extreme climate, and forest ownership [75–79].

Added references

  • Korea Forest Service Forest Basic Statistics for 2020; Daejeon, 2021
  • Kwon, K., Han, H., Seol, A., Chung, H., & Chung, J. (2016). Analyzing thinning effects on growth and carbon absorption for Cryptomeria japonica stands using distance-independent growth simulations. Journal of Korean Society of Forest Science, 105(1), 132-138.
  • Cortini, F., & Comeau, P. G. (2020). Pests, climate and competition effects on survival and growth of trembling aspen in western Canada. New Forests, 51(1), 175-190.
  • Jandl, R., Ledermann, T., Kindermann, G., Freudenschuss, A., Gschwantner, T., & Weiss, P. (2018). Strategies for climate-smart forest management in Austria. Forests, 9(10), 592.
  • Kim, K. D. (2016). Analysis of decision factors on the participation of scaling project for private forest management using a Logit model. Journal of Korean Society of Forest Science, 105(3), 360-365.
  • Woo, H., Han, H., Cho, S., Jung, G., Kim, B., Ryu, J., ... & Park, J. (2020). Investigating the optimal location of potential forest industry clusters to enhance domestic timber utilization in South Korea. Forests, 11(9), 936.
  • Ficko, A., Lidestav, G., Dhubháin, Á. N., Karppinen, H., Zivojinovic, I., & Westin, K. (2019). European private forest owner typologies: A review of methods and use. Forest Policy and Economics, 99, 21-31.

Comment 6

Also suggested adding one specific para on the risk management options including FLR, NbS to justify your research Forest and Landscape Restoration (FLR)

→Response

Thank you for your valuable comment on our manuscript. According to your comment, we have added sentences and references in Methodology and Discussion section. According to Bae et al., 2012, the forests of South Korea were successfully restored through continuous forest restoration projects after being devastated in the aftermath of the war. corresponds. Furthermore, scenario 4 based on forest policy including future climate change response and adaptation includes NbS. Therefore, the results and considerations from that perspective were added to this manuscript.

(P7 L219-225)

Scenario 3 and 4 were planned to simulate changes in forest physiognomy according to the degree of the adaptive pathway. Furthermore, Scenario 3 was constructed to analyze the aspects of Forest and Landscape Restoration (FLR), and Scenario 4 to Nature-based Solutions (NBS) [63,64].

Added references

  • Bae, J. S., Joo, R. W., & Kim, Y. S. (2012). Forest transition in South Korea: reality, path and drivers. Land use policy, 29(1), 198-207.
  • Kim, G., Kim, J., Ko, Y., Eyman, O. T. G., Chowdhury, S., Adiwal, J., ... & Son, Y. (2021). How Do Nature-Based Solutions Improve Environmental and Socio-Economic Resilience to Achieve the Sustainable Development Goals? Reforestation and Afforestation Cases from the Republic of Korea. Sustainability, 13(21), 12171.

(P17 L451-454)

Therefore, it was estimated that the more the forest management progressed, the smaller the vulnerable area and the lower the risk grade compared to the overprotected scenario. This suggests that forest management including NBS as well as FLR should be more activated.

Reviewer 4 Report

1. Any model can simulate and predict. Is it worth believing? Simulation work must follow the steps of simulation research (https://uh.edu/~lcr3600/simulation/steps.html). In addition to defining the model and analyzing the model results, the calibration-evaluation and implication of the model are important. Unfortunately, I did not see the relevant content, which means that the results are not evaluated and unreliable. It is important to deal with model uncertainty

2. The whole manuscript only stays at the level of concept definition, data and simulation results; there is a lack of review on relevant theories, top models, as well as leader scientists in the field. Discussions repeated results without novel points. Introduction and discussion sections need an international perspective. This means that it is only a local study and does not attract international readers.

Author Response

Answers to Reviewers’ Comments

Title: Risk Assessment of Climate Change in Forest based on Growth Modeling in South Korea.

We appreciate your valuable comments and suggestions; these have helped us to improve upon the quality of our manuscript. The manuscript has been revised in accordance with your suggestions. The revised manuscript has been re-checked by a native English language editor to improve readability. Please find our responses below for each comment (P and L mean page number and line number in “Revised manuscript”). We hope that you find our revised manuscript acceptable.

Comment 1

Any model can simulate and predict. Is it worth believing? Simulation work must follow the steps of simulation research (https://uh.edu/~lcr3600/simulation/steps.html). In addition to defining the model and analyzing the model results, the calibration-evaluation and implication of the model are important. Unfortunately, I did not see the relevant content, which means that the results are not evaluated and unreliable. It is important to deal with model uncertainty

→Response

Thank you for your kind advice and we agreed with your comment. We have added L192-199 and references. In the previous study by Kim et al., 2019, this model was calibrated through the 5th and 6th NFI and verified through the national forestry statistics and other research (R2 = 0.96, RMSE = 9.13).

(P7 L192-199)

It can enhance the model's performance, reducing uncertainty about future estimates due to climate change as it reflects Korea's ecological and environmental characteristics. This can be used to derive growing stock using information such as diameter at breast height, tree height, and stand density [60]. Furthermore, the KO-G-Dynamic model was com-pared and calibrated through the fifth and sixth National Forest Inventory (NFI) Data and verified through national forestry statistical data and other research [55].

Added references

  • Kim, M., Kraxner, F., Son, Y., Jeon, S. W., Shvidenko, A., Schepaschenko, D., ... & Lee, W. K. (2019). Quantifying impacts of national-scale afforestation on carbon budgets in South Korea from 1961 to 2014. Forests, 10(7), 579.
  • Hong, M., Song, C., Kim, M., Kim, J., Lee, S. G., Lim, C. H., ... & Lee, W. K. (2022). Application of integrated Korean forest growth dynamics model to meet NDC target by considering forest management scenarios and budget. Carbon Balance and Management, 17(1), 1-18.

Comment 2

The whole manuscript only stays at the level of concept definition, data and simulation results; there is a lack of review on relevant theories, top models, as well as leader scientists in the field. Discussions repeated results without novel points. Introduction and discussion sections need an international perspective. This means that it is only a local study and does not attract international readers.

→Response

Thank you for your comment. In this study, As the IPCC AR is updated, the contents of the defined concept assessment change are summarized.

In the forest growth risk assessment, it is meaningful to evaluate it as climate factors accumulate and affect growth. Although there has been much research on forest assessment in the past, research on evaluation through linkages between factors still was insufficient. Therefore, this study analyzed risk assessment through the mechanical linkage between factors, which is meaningful in that policy priorities can be derived through decision-making.

In addition, although the study analyzed a pilot study on forest growth in South Korea, the tendency of growth to decrease abroad is similar due to the nature of forests. Therefore, this study can be helpful to countries trying to establish adaptation measures or attempt risk assessments.

(P17 L430-431)

Assessing the risk of the forest sector due to climate change in the Republic of Korea will help the country recognize the hazards of forest growth and establish adaptive measures [69]. There were many assessments in previous studies, but studies on the linkage between assessment factors were insufficient [52,70,71]. In this study, the vulnerability and growth-related risks were evaluated to fit the AR5 and AR6 frames based on a vali-dated model with the qualitative and limited adjustment range of the vulnerability assessment covered by AR4.

Added references

  • Cui, G., Kwak, H., Choi, S., Kim, M., Lim, C. H., Lee, W. K., ... & Chae, Y. (2016). Assessing vulnerability of forests to climate change in South Korea. Journal of Forestry Research, 27, 489-503.
  • Reinmann, A. B., Susser, J. R., Demaria, E. M., & Templer, P. H. (2019). Declines in northern forest tree growth following snowpack decline and soil freezing. Global Change Biology, 25(2), 420-430.
  • Pokhriyal, P., Rehman, S., Areendran, G., Raj, K., Pandey, R., Kumar, M., ... & Sajjad, H. (2020). Assessing forest cover vulnerability in Uttarakhand, India using analytical hierarchy process. Modeling Earth Systems and Environment, 6, 821-831.

(P17 L478-484)

This suggests an optimal climate-resilient development pathway for sustainable forests. Furthermore, this study can contribute to establishing carbon-neutral and sustainable policy goals in the rapidly changing climate and socioeconomic environment. In addition, as this study confirms that risk assessment of the forest growth sector in South Korea is possible, it suggests that it can be applied to risk assessment of the overseas forest growth sector in the future. Also, it can support other countries in establishing international adaptation policies or implementing risk assessments.

Round 2

Reviewer 1 Report

The authors made necessary revision on the manuscript. I think the current one can be accepted for publication.

Author Response

Thanks for your kind comments. Thank you for taking the valuable time to review.

We additionally reflected on the opinions of other minor reviewers.

Reviewer 4 Report

I feel that the review of methods section can be strengthened.

The introduction to the ko-g-dynamic model is too simple, and the independence of this manuscript will be questioned. The manuscript needs to carefully describe the model, algorithm, process and evaluation results, so that readers can understand the model without reading 55 references.

Globally, gozens of algorithms and methods have been developed for climate change vulnerability and climate change risk assessment for various ecosystems, including forests. I didn't see a depth review corresponding contents in the introduction, nor did they were commented in the discussion.

Author Response

Answers to Reviewers’ Comments

Title: Risk Assessment of Climate Change in Forest based on Growth Modeling in South Korea.

We appreciate your valuable comments and suggestions; these have helped us to improve upon the quality of our manuscript. The manuscript has been revised in accordance with your suggestions. The revised manuscript has been re-checked by a native English language editor to improve readability. Please find our responses below for each comment (P and L mean page number and line number in “Revised manuscript”). We hope that you find our revised manuscript acceptable.

Comment 1

The introduction to the ko-g-dynamic model is too simple, and the independence of this manuscript will be questioned. The manuscript needs to carefully describe the model, algorithm, process and evaluation results, so that readers can understand the model without reading 55 references.

→Response

Thank you for your valuable comment on our manuscript. According to your comment, we added contents that the KO-G-Dynamic model's process through the model's algorithm and coefficients and the suitability of South Korea.

(P7 L191-207)

The 'KO-G-Dynamic model,' a forest growth model, was used to apply risk assessment modeling techniques in this study [57]. Using the KO-G-Dynamic model has enhanced the forest growth of previous studies [58–61]. We constructed the forest stands in 1 km × 1 km grids and described the state of each grid with four variables with varying values: forest type, site index, stand age, and management type using the fifth NFI and the fifth Forest Cover Map. In addition, each grid was spatially linked to climatic and topographic conditions from digital elevation model data and climate data [62]. Coefficients determined for each tree species, were incorporated with the biomass allometric equation data developed by the National Institute of Forest Science (NIFoS) [63]. It can predict the annual growth of temperate forest more accurately, including the climate impact overlooked by traditional dynamic growth models. It can enhance the model's performance, reducing uncertainty about future estimates due to climate change as it reflects Korea's ecological and environmental characteristics. This can be used to derive growing stock using information such as diameter at breast height, tree height, and stand density [64]. Furthermore, the KO-G-Dynamic model was compared and calibrated through the fifth and sixth National Forest Inventory (NFI) Data and verified through national forestry statistical data and other research [57].

Added references

  • Son YM, Lee GH, Kim R, Pyo JK, Park IH, Son YW, Kim C. Carbon emission factors by major tree species for forest greenhouse gas inventory. Seoul; 2010.
  • Kim, M., Kraxner, F., Forsell, N., Song, C., & Lee, W. K. (2021). Enhancing the provisioning of ecosystem services in South Korea under climate change: The benefits and pitfalls of current forest management strategies. Regional Environmental Change, 21, 1-10.

Comment 2

Globally, dozens of algorithms and methods have been developed for climate change vulnerability and climate change risk assessment for various ecosystems, including forests. I didn't see a depth review corresponding contents in the introduction, nor did they were commented in the discussion.

→Response

Thank you for your advice and suggestion. Based on your comment, the contents of the assessment using the global model were added, and the necessity of this study was added to the introduction and discussion.

(P2 L92-96)

This means Korean forests require proper management and climate change predictions [15]. Forests are also important global carbon sinks [16]. Previous studies evaluated South Korea's forest ecosystem through global models [17–19]. However, mechanical and numerical risk assessments have not comprehensively considered forest management plans and practices in South Korea. Therefore, assessing climate change risk requires proper spatiotemporal models that reflect conceptual linkages among criteria.

Added references

  • Choi, S., Lee, W. K., Kwak, H., Kim, S. R., Yoo, S., Choi, H. A., ... & Lim, J. H. (2011). Vulnerability assessment of forest ecosystem to climate change in Korea using MC1 model). Journal of Forest Planning, 16(Special_Issue), 149-161.
  • Song, C., Pietsch, S. A., Kim, M., Cha, S., Park, E., Shvidenko, A., ... & Lee, W. K. (2019). Assessing forest ecosystems across the vertical edge of the mid-latitude ecotone using the biogeochemistry management model (BGC-MAN). Forests, 10(6), 523.
  • Choi, Y., Lim, C. H., Krasovskiy, A., Platov, A., Kim, Y., Chung, H. I., ... & Jeon, S. W. (2022). Can a national afforestation plan achieve simultaneous goals of biodiversity and carbon enhancement? Exploring optimal decision making using multi-spatial modeling. Biological Conservation, 267, 109474.

(P19 L491-492)

This suggests an optimal climate-resilient development pathway for sustainable forests. Furthermore, this study can contribute to establishing carbon-neutral and sustainable policy goals in the rapidly changing climate and socioeconomic environment. In addition, as this study confirms that risk assessment of the forest growth sector in South Korea is possible, it suggests that it can be applied to risk assessment of the overseas forest growth sector in the future. Also, it can support other countries in establishing international adaptation policies or implementing risk assessments.